# Fight Against Antimicrobial Resistance: We Always Need New Antibacterials but for Right Bacteria

**DOI:** 10.3390/molecules24173152

**Published:** 2019-08-29

**Authors:** Raphaël E. Duval, Marion Grare, Béatrice Demoré

**Affiliations:** 1Université de Lorraine, CNRS, L2CM, F-54000 Nancy, France; 2ABC Platform®, Faculté de Pharmacie, F-54505 Vandœuvre-lès-Nancy, France; 3Laboratoire de Bactériologie-Hygiène, CHU Toulouse, F-31059 Toulouse, France; 4CHRU de Nancy, pôle Pharmacie-Stérilisation, F-54511 Vandœuvre-lès-Nancy, France; 5Université de Lorraine, APEMAC, F-54000 Nancy, France

**Keywords:** antimicrobial resistance, multidrug-resistant bacteria, drug discovery, new antibacterials

## Abstract

Antimicrobial resistance in bacteria is frightening, especially resistance in Gram-negative Bacteria (GNB). In 2017, the World Health Organization (WHO) published a list of 12 bacteria that represent a threat to human health, and among these, a majority of GNB. Antibiotic resistance is a complex and relatively old phenomenon that is the consequence of several factors. The first factor is the vertiginous drop in research and development of new antibacterials. In fact, many companies simply stop this R&D activity. The finding is simple: there are enough antibiotics to treat the different types of infection that clinicians face. The second factor is the appearance and spread of resistant or even multidrug-resistant bacteria. For a long time, this situation remained rather confidential, almost anecdotal. It was not until the end of the 1980s that awareness emerged. It was the time of Vancomycin-Resistance Enterococci (VRE), and the threat of Vancomycin-Resistant MRSA (Methicillin-Resistant *Staphylococcus*
*aureus*). After this, there has been renewed interest but only in anti-Gram positive antibacterials. Today, the threat is GNB, and we have no new molecules with innovative mechanism of action to fight effectively against these bugs. However, the war against antimicrobial resistance is not lost. We must continue the fight, which requires a better knowledge of the mechanisms of action of anti-infectious agents and concomitantly the mechanisms of resistance of infectious agents.

## 1. Introduction

In 2004, faced with the lack of new antibacterial molecules and the withdrawal of the pharmaceutical industry in the search for new antibiotics, the Infectious Disease Society of America (IDSA) published the report “Bad Bugs, No Drugs” warning of the risks incurred in terms of cost to society (about 5 billion dollars a year, only in the US) and a future public health crisis of great concern [1,2]. The following year, IDSA launched the program “Bad Bugs, Need Drugs: 10 × ‘20 Initiative” to boost antibiotic research with the objective to have 10 new systemic antibiotics by 2020 [3].

This review aims to take stock, 10 years after the initiative taken by IDSA, on: (i) multidrug-resistant bacteria that are of particular concern and for which it is urgent to find new molecules, (ii) antimicrobial resistance and the risk incurred of a major public health crisis if new molecules are not available, and (iii) the search for new antibacterial molecules, the status of research, and the adequacy with the bacteria to be targeted.

## 2. What is the Threat?

On Monday, 27 February 2017, the World Health Organization (WHO) published on its website a list of 12 bacteria whose level of resistance to antibiotics is such that they represent a real threat to human health; and that it is now urgent to act [4]. The WHO has divided these 12 bacteria into 3 groups according to their urgency (Table 1):

It is striking that this list particularly highlights the threat posed by Gram-negative bacteria (GNB), and points more specifically to Enterobacteriaceae. In fact, of the 12 bacteria, 9 are GNBs: *Acinetobacter baumannii*, *Pseudomonas aeruginosa*, Enterobacteriaceae, *Helicobacter pylori*, *Campylobacter* spp., *Salmonella* spp., *Neisseria gonorrhoeae*, *Haemophilus influenzae*, *Shigella* spp.; and of these 9, at least 3 belong to the Enterobacteriaceae family. Indeed, if we consider the group “Enterobacteriaceae, carbapenem-resistant, 3rd generation cephalosporin-resistant” it should be understood that the bacteria: *Klebsiella pneumoniae*, *Escherichia coli*, *Enterobacter* spp., *Serratia* spp., and *Proteus* spp. are included in this group (Table 1).

*De facto*, among multidrug-resistant bacteria, GNBs are the bacteria most frequently encountered during infections in humans. Indeed, in a very recent published review, a team of European researchers associated with the ECDC (European Centre for Disease Prevention and Control) estimated that GNBs are responsible for more than 500,000 infections (out of a total of 672,000) and more than 24,600 deaths (out of a total of 33,000) in only one year (i.e., 2015) in Europe [6]. In the end, the WHO press release highlights the real threat posed by GNBs who have developed remarkable skills enabling them to “find new ways to resist treatments and (to) transmit genetic material allowing other bacteria to become resistant as well” [4]. In conclusion, among its proposals for combating so-called antimicrobial resistance, WHO stresses the need to intensify research and development on new antibiotics; but at the same time, it states that R&D will not solve alone the problem of resistance or multidrug resistance to antibiotics in bacteria.

A few months earlier, in 2016, the report “Tackling Drug-Resistant Infections Globally: Final Report and Recommendations” was handed over to Government of the United Kingdom [7]. This report was the result of an extensive work realized by the independent commission “Review on Antimicrobial Resistance”, a group of experts created and supported by the joint initiative of the Wellcome Trust and UK Department of Health. The tasks of this working group were (i) to analyze from a global point of view the increase in resistance to anti-infectives and (ii) to propose concrete actions to face antimicrobial resistance worldwide [8]. One of the main data that emerges from this report is that by 2050, the number of deaths attributable to resistance to anti-infective drugs will approach the number of 10 million people per year; more than road accidents, diabetes, and even cancer (Figure 1), the equivalent of the population of countries like Sweden, the Czech Republic, Greece, or Portugal [9].

However, this report has not been limited to a disturbing projection. It has also been a force of proposal and has identified 10 major actions (or “Ten Fronts”) to fight against antimicrobial resistance [7].

These 10 actions are:Massive and global public awareness campaign;Improve hygiene and prevent the spread of infections;Reduce the (unnecessary) use of antimicrobials in agriculture and their dissemination in the environment;Improve global surveillance for (i) antimicrobial resistance and (ii) antimicrobial consumption in humans and animals;Promote new and rapid diagnostic tests to stop the use of antibiotics as quickly as possible;Promote the development and use of vaccines and (therapeutic) alternatives;Improve the number, remuneration, and recognition of people working in infectious diseases;Establish a global innovation fund for pre-clinical and non-commercial research;Promote better investment for new medicines and improve existing ones;Build a global coalition for real action - via the G20 and the UN.

It is striking that the fight against antimicrobial resistance is not limited to only one field in Health, or only one field in Research. The fight against antimicrobial resistance is everyone’s business. We are all concerned and all responsible not to be one day all affected by antimicrobial resistance.

## 3. How did We Get There?

Antibiotic resistance is a complex and relatively old phenomenon that is the consequence of several factors.

The first factor is the drop-in research and development of new antibacterials. Indeed, we can see that since the beginning of the 1980s, the number of new antibiotics put on the market has been steadily decreasing (Figure 2) to reach the critical number of only 2 specialties approved by the FDA for the period 2008–2012.

In the early 1980s, R&D on antibiotics culminated; virtually all pharmaceutical companies are active in the field of antibiotic research. Therefore, we were faced with a paradox: there were many new antibiotics, but in reality, there were too many. Hence, it was difficult for a pharmaceutical industry to launch new molecules on the market, and above all they were profitable. As a result, low investment returns, coupled with unpredictable and often impractical approval procedures put in place by the agencies (i.e., FDA and EMA) have pushed many companies out of the antibiotic market.

Unfortunately, this phenomenon will be amplified by two events:

During the first event, from the years 1970–1980, one begins to see redemptions, mergers, absorptions, etc., become widespread within the chemical/pharmaceutical companies, with the creation of “big pharma” to face the competition. However, this will also have the consequence of accelerating the closure of the R&D department and the abandonment of molecules under development… These R&D departments will even tend to disappear from pharmaceutical companies that will outsource this R&D activity to start-ups. Indeed, the R&D department was considered too risky and especially not profitable quickly enough. It requires a lot of resources and in general the time between discovery and marketing of a new antibiotic is estimated to be 10 years. This movement of disengagement of pharmaceutical industries in the field of antibiotic research will continue to accelerate from the 1990s [13,14].

The second event will directly “boost” the disengagement of pharmaceutical industries. It is linked to the advent of a radical change in the policy for management of antibiotic therapy at the level of health facilities: the rationalization of the prescription of antibiotics in order to achieve a fair use of antibiotics.

The second factor is the appearance and spread of resistant or even multidrug-resistant bacteria. If we look at Figure 3, we can see the ease, and especially the speed with which bacteria are able to become resistant to an antibiotic after its discovery and introduction clinic. Overall, it takes less than 10 years to a bacterium to become potentially resistant to an antibiotic.

For a long time, this situation remained rather confidential, almost anecdotal. It was not until the end of the 1980s that an awareness emerged. It was a consequence of the appearance of glycopeptide (mainly Vancomycin) resistance in Enterococci [15]. Indeed, this mechanism of resistance is (predominantly) carried by a plasmid, and the risk was related to its possible transfer to Staphylococci: there would be no more therapeutic alternative for the management of Methicillin-Resistant *Staphylococcus aureus* infections which already were a source of concern, mainly in the United States. This was the first case of antimicrobial resistance for which there has been a real concern and therefore an awareness: if Staphylococci became resistant to Vancomycin (and by extension to Glycopeptides) it would be the beginning of the end.

This event is also important for another reason. It is the source of an error in the fight against antimicrobial resistance: focus Research on the discovery of new antibacterials with a very narrow spectrum; here in this case anti-staphylococcal molecules.

Indeed, if we look at Table 2, we can see that since 1998, there is still a significant number of new broad-spectrum antibiotics; but at the same time, molecules with a narrow spectrum are almost exclusively molecules directed against Gram-positive bacteria (GPB), not to say only anti-staphylococcal, creating a clear imbalance.

So, we are faced with a worrying situation: GNBs are advertised as a threat, they are becoming more resistant, multidrug-resistant GNB infections are more common, and we have limited therapeutic options to fight against these bacteria.

This absolute emergency has forced clinicians to turn to old molecules, such as colistin, to treat multidrug-resistant GNB infections; allowing, at least for a time, to continue to take care of patients [16]. Colistin was then called a last-chance (or a last-resort) antibiotic [17].Unfortunately, and very quickly, the first colistin-resistant GNBs appeared all around the world, including in Europe [18,19,20,21,22,23]. This evolution led to us fearing the worst: a “post-antibiotic era” with, in the end, the real risk of not being able to take care of patients suffering from bacterial infections” [24], which led WHO, from 2014, to emit a first warning signal and declare that it was urgent to act. [25].

## 4. Is the War Lost?

Until the beginning of the 2010s, there were very few new molecules and in particular few new molecules with a spectrum directed against GNBs, but the situation seems to be improving in recent years (Figure 2).

Indeed, very recently, new specialties meeting these criteria have appeared on the market: ceftolozane/tazobactam, ceftazidime/avibactam, and meropenem/vaborbactam (Table 2):✓ceftolozane (3rd generation cephalosporin) + tazobactam (β-lactamase inhibitor): combination which has a rather broad antibacterial spectrum: *Pseudomonas aeruginosa*, Enterobacteriaceae responsible for community infections (*Escherichia coli*, *Proteus mirabilis*, *Proteus vulgaris*, *Salmonella* spp.) and Enterobacteriaceae producing cephalosporinases responsible for nosocomial infections (*Citrobacter freundii*, *Morganella morganii*, and *Serratia marcescens*). On the other hand, this combination is less active on extended-spectrum β-lactamase-producing (ESBL) *Klebsiella pneumoniae* strains and ceftazidime-resistant *Enterobacter* spp. strains (i.e., the strains which overexpressed AmpC) [26].✓ceftazidime (3rd generation cephalosporin) + avibactam (non-β-lactam β-lactamase inhibitor): avibactam is a β-lactamase inhibitor of a new class (i.e., diazabicyclooctanones) which has a broader inhibitory activity than “classical” β-lactamase inhibitors. It inhibits both class A and class C enzymes (i.e., Ambler classification), including extended spectrum β-lactamases (ESBL), KPC and OXA-48 carbapenemases, and AmpC enzymes. However, it has no effect on class B enzymes (metallo-β-lactamases) and is not capable of inhibiting many class D enzymes [27].✓meropenem (carbapenem) + vaborbactam (non-β-lactam β-lactamase inhibitor): vaborbactam, a β-lactamase inhibitor of a new class (cyclic boronates), prevents certain classes of β-lactamases (class A and class C) from hydrolyzing meropenem and therefore restores its activity in many infections due to carbapenem-resistant Enterobacteriaceae. [28].

These combinations represent very interesting new therapeutic options, but they are not, in any case, new molecules, *sensus stricto*. Indeed, we must remain cautious because these antibiotics belong to the β-lactam family. Moreover, ceftazidime, is even an “old” molecule, used in therapy since 1985. Unfortunately, resistance to the first two associations have already been observed [29,30].

In the end, it is likely that if we do not bring innovation and discover new antibiotic molecules (i.e., new chemical structures), ideally with a new mechanism of action (e.g., the last antibiotic approved by the FDA belongs to the family of cyclines (Table 2)), the fight against antimicrobial resistance may be long and very difficult. Indeed, a study recently published by the ECDC demonstrated that infections caused by antibiotic-resistant bacteria were responsible for more than 33,000 deaths in 2015, only in Europe [6].

## 5. Conclusions

Antimicrobial resistance (AMR), and more specifically, antibiotic resistance, has become a particularly serious public health problem for several years. However, AMR is not a new phenomenon: bacteria have always been able to adapt, develop, or acquire mechanisms of resistance to antibiotics. Many reports and publications try to alert us to the need we have to react [31]. One of the priorities of research to face AMR is the search for new antibacterial molecules, even if is not the only one. Indeed, this situation has become even more critical as the therapeutic arsenal is shrinking and we are facing a glaring lack of new molecules (i.e., with new chemical structures or new mechanisms of action). Moreover, it is clear that we urgently need new molecules to fight Gram-negative bacteria that currently pose the greatest threat.

In any case, the war against antimicrobial resistance is not lost. We must continue the fight, which requires a better knowledge of the mechanisms of action of anti-infectious agents and concomitantly the mechanisms of resistance of infectious agents, a better and fairer use of antibiotics, change in mentalities compared to antibiotics, and above all, promote the R&D of new anti-infectives. We need “real” new molecules, i.e., with new chemical structures. However, perhaps we finally see the end of the tunnel. Indeed, on 19 August 2019, the FDA approved the marketing of a brand-new antibiotic: Lefamulin, for the treatment of community-acquired bacterial-pneumonia (CABP) [32]. Lefamulin is the first antibiotic with a novel mechanism of action approved by the FDA in nearly 20 years. Lefamulin belongs to the family of pleuromutilins, which targets a different protein synthesis binding site than older antibiotics (i.e., it inhibits of protein synthesis by binding to the peptidyl transferase center of the 50S bacterial ribosome, thus preventing the binding of transfer RNA for peptide transfer) [33]. Lefamulin represents a great hope in the search for real new antibacterials. We must therefore continue to search for new antibacterial molecules and synthetic or natural molecules to feed the pipeline, because we will never stop needing new antibacterials.

## Figures and Tables

**Figure 1 molecules-24-03152-f001:**
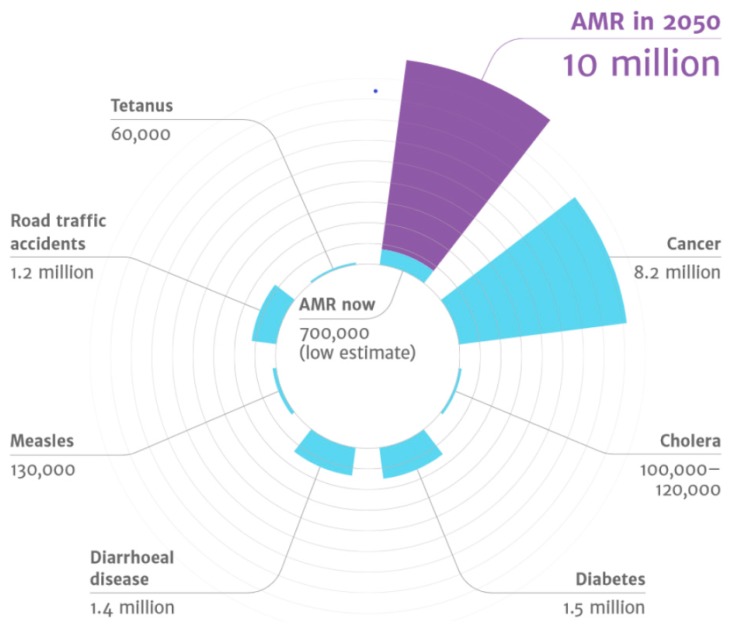
The main causes of death by 2050 [7].

**Figure 2 molecules-24-03152-f002:**
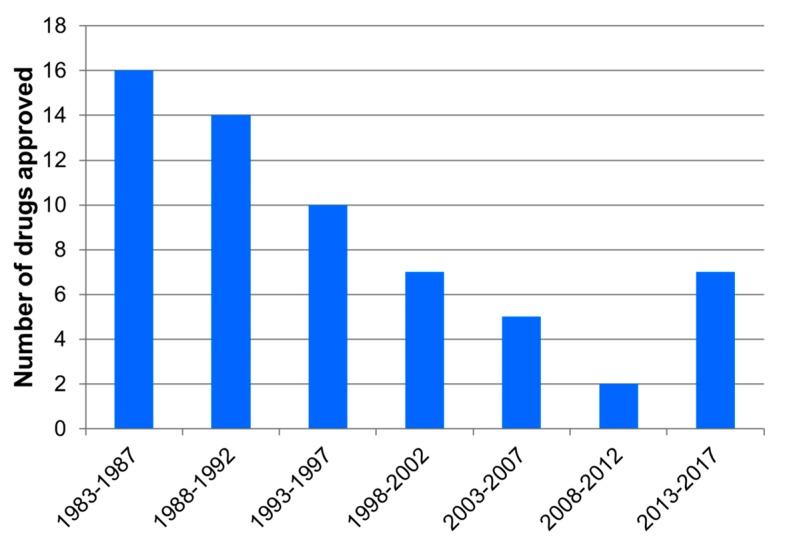
Evolution of the FDA-approved antibiotics since 1983. Modified from Reference [10] and completed with References [11,12].

**Figure 3 molecules-24-03152-f003:**
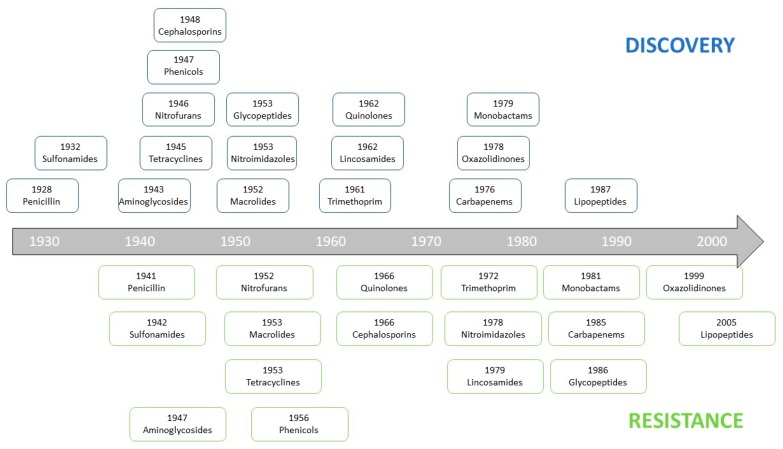
Antibiotics timeline from the end of the 1920s until today, indicating when the main antibiotic classes were discovered, and when the mechanisms of resistance to these antibiotics were first described.

**Table 1 molecules-24-03152-t001:** WHO priority pathogens list for research and development (R&D) of new antibiotics [5].

**Priority 1: Urgency “Critical” #**
***Acinetobacter baumannii***, carbapenem-resistant
***Pseudomonas aeruginosa***, carbapenem-resistant ^§^
**Enterobacteriaceae** *, carbapenem-resistant, 3rd generation cephalosporin-resistant
**Priority 2: Urgency “High”**
***Enterococcus faecium***, vancomycin-resistant
***Staphylococcus aureus***, methicillin-resistant, vancomycin intermediate and resistant
***Helicobacter pylori***, clarithromycin-resistant
***Campylobacter* spp.**, fluoroquinolone-resistant
***Salmonella* spp.**, fluoroquinolone-resistant
***Neisseria gonorrhoeae***, 3rd generation cephalosporin-resistant, fluoroquinolone-resistant
**Priority 3: Urgency “Medium”**
***Streptococcus pneumoniae***, penicillin-non-susceptible
***Haemophilus influenzae***, ampicillin-resistant
***Shigella* spp.**, fluoroquinolone-resistant

#: *Mycobacteria* (including *Mycobacterium tuberculosis*, the cause of human tuberculosis), was not subjected to review for inclusion in this prioritization exercise as it is already a globally established priority for which innovative new treatments are urgently needed. ^§^: *Pseudomonas aeruginosa* carbapenem-resistant = resistant to all β-lactams (strains only resistant to imipenem by porin D2 modification and/or only resistant to meropenem by efflux, are not concerned). *: Enterobacteriaceae include: *Klebsiella pneumoniae*, *Escherichia coli*, *Enterobacter* spp., *Serratia* spp., *Proteus* spp., and *Providencia* spp., *Morganella* spp.

**Table 2 molecules-24-03152-t002:** List of systemic antibiotics approved by the FDA and EMA since 1999.

Antibacterial	Year Approved		Novel Mechanism?	Spectra
	FDA	EMA		
Quinupristin/dalfopristin	1999	2000	No	GPB
Moxifloxacin	1999	2001	No	GPB-GNB
Gatifloxacin *	1999	/	No	GPB-GNB
Linezolid	2000	2001	Yes	GPB
Cefditoren pivoxil	2001	/	No	GPB-GNB
Ertapenem	2001	2002	No	GNB-GPB
Gemifloxacin *	2003	/	No	GPB-GNB
Daptomycin	2003	2006	Yes	GPB
Telithromycin *	2004	2001	No	GPB
Tigecycline	2005	2006	Yes	GPB-GNB
Doripenem *	2007	2008	No	GNB-GPB
Telavancin	2009	2011	Yes	GPB
Ceftarolin fosamil	2010	2012	No	GPB-GNB
Ceftolozane-tazobactam	2014	2015	No	GNB-GPB
Tedizolid	2014	2015	No	GPB
Oritavancin	2014	2015	No	GPB
Dalbavancin	2014	2015	No	GPB
Ceftazidime-avibactam	2015	2016	No	GNB
Meropenem-vaborbactam	2017	2018	No	GPB-GNB
Delafloxacin	2017	/	No	GPB-GNB
Omadacycline	2018	/	No	GPB-GNB

Modified from [10] and completed with [11,12]. FDA: Food and Drug Administration; EMA: European Medicines Agency; *: withdraw from the market; /: not approved, so far, by EMA; GPB: Gram-positive bacteria; GNB: Gram-negative bacteria.

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
