# Peer review of "Fight Against Antimicrobial Resistance: We Always Need New Antibacterials but for Right Bacteria"

_molecules, 2019, doi:10.3390/molecules24173152_

Round 1

Reviewer 1 Report

The manuscript prepared by Raphaёl E. Duval and co- workers described very important problem – multiresistance in bacteria Gram-negative (GNB), to confirm the above problem, the authors refer to data presented by the WHO.The authors present - Antibiotics resistance is a complex and relatively old phenomen that is the consequence of several factors.  

I am posting my comments below:

1.      in the Abstract  section the authors present problem GNB  but do not provide literature support for this reference - this should be supplemented by the authors 

the manuscript does not contain introduction and conclusion ?????

 3.      in my opinion, the presented problem requires a summary in the presented manuscript, the lack of such a summary of the performed drug resistance analysis

After careful reading of this paper I am of the opinion that the work is suitable to Molecules after making corrections and additions

Author Response

Reviewer 1

The manuscript prepared by Raphaёl E. Duval and co- workers described very important problem – multiresistance in bacteria Gram-negative (GNB), to confirm the above problem, the authors refer to data presented by the WHO. The authors present - Antibiotics resistance is a complex and relatively old phenomen that is the consequence of several factors.

Dear reviewer

Thank you for all the remarks you made, and which allowed us to offer you the revised manuscript attached. All changes made to the manuscript are highlighted in yellow.

I am posting my comments below:

in the Abstract section the authors present problem GNB but do not provide literature support for this reference - this should be supplemented by the authors

In the original version of the manuscript, we included reference [26]

Cassini, A.; Högberg, L.D.; Plachouras, D.; Quattrocchi, A.; Hoxha, A.; Simonsen, G.S.; Colomb-Cotinat, M.; Kretzschmar, M.E.; Devleesschauwer, B.; Cecchini, M.; Ouakrim, D.A.; Oliveira, T.C.; Struelens, M.J.; Suetens, C.; Monnet, D.L.; Burden of AMR Collaborative Group. Attributable deaths and disability-adjusted life-years caused by infections with antibiotic-resistant bacteria in the EU and the European Economic Area in 2015: a population-level modelling analysis. Lancet Infect Dis. 2019, 19, 56-66. doi: 10.1016/S1473-3099(18)30605-4.

If you look at the Table 1 “Estimated annual burden of infection with antibiotic-resistant bacteria of public health importance, by decreasing number of DALYs per 100 0000 population, EU and European Economic Area, 2015” of this reference, you can see and calculate that GNB are responsible of more than 500,000 infections (out of a total of 672,000) and more than 24,600 deaths (out of a total of 33,000) in only one year (i.e. 2015) in Europe.

However, these data may not be sufficiently visible in the previous version of the manuscript, so we have inserted this data into this modified manuscript as follow:

“Indeed, in a very recent published review, a team of European researchers associated with the ECDC (European Centre for Disease Prevention and Control) estimated that GNBs are responsible of more than 500,000 infections (out of a total of 672,000) and more than 24,600 deaths (out of a total of 33,000) in only one year (i.e. 2015) in Europe [6].

the manuscript does not contain introduction and conclusion?????

Our initial intention was to propose a manuscript of the type "communication" or "comment" which can explain the original structure of the manuscript: absence of an introduction and a conclusion, with separate paragraphs ... We have modified this point ...

in my opinion, the presented problem requires a summary in the presented manuscript, the lack of such a summary of the performed drug resistance analysis

We are really sorry, but we do not understand the correction requested by the reviewer 1.

the presented problem requires a summary in the presented manuscript” what are we talking about? Which summary? We are sorry, but we do not understand …

“the lack of such a summary of the performed drug resistance analysis”…? Again, we are sorry, but we do not understand this sentence.

Reviewer 2 Report

Dear Authors:

While the topic is relevant and is a topic to be taken into account above all for scientists seeking solutions in the generation and formulation of antibiotics, the information is not new and is nothing more than a brochure or a more extensive newspaper news, and It does not go with the lines of the magazine.

on the other hand there are some mistakes to correct:

-line 1 whitout parentheses

-To give a logical order to the manuscript without introduction, .... and conclusion

-line 36 and 53 Table I changed by Table 1

-In table 1 all spp. whitout italics

- redesign the table very confusing and messy

-Line 47 and 55, pag 2, Enterobacteriaceae whitout italics

-Line 62, pag, incomplete

-Line 163 Staphylococci whitout italics

Author Response

Reviewer 2

Dear Authors:

While the topic is relevant and is a topic to be taken into account above all for scientists seeking solutions in the generation and formulation of antibiotics, the information is not new and is nothing more than a brochure or a more extensive newspaper news, and It does not go with the lines of the magazine.

Dear reviewer

Thank you for all the remarks you made, and which allowed us to offer you the revised manuscript attached.

All changes made to the manuscript are highlighted in yellow.

on the other hand there are some mistakes to correct:

-line 1 whitout parentheses

The title of the manuscript has been changed as requested

-To give a logical order to the manuscript without introduction, .... and conclusion

Initially we wanted to submit to Molecules our manuscript as a communication ... At the request of the Academic Editor we submit you a short review ... so we added an introduction and a conclusion.

-line 36 and 53 Table I changed by Table 1

The corrections were made as requested.

-In table 1 all spp. whitout italics

The corrections were made as requested.

- redesign the table very confusing and messy

As requested, we propose you a new presentation of Table 1, we hope you find it less confusing and messy.

-Line 47 and 55, pag 2, Enterobacteriaceae whitout italics

The corrections were made as requested. As you will see, we also checked the entire manuscript ... there were other errors of this type ...

-Line 62, pag, incomplete

We are really sorry, but we do not understand the correction requested by the reviewer 2.

-Line 163 Staphylococci whitout italics

The corrections were made as requested.

Reviewer 3 Report

Well done, indeed. I really liked Your valuable manuscript (MS). My sincere congrats to all authors. 

Overall Recommendation: Accept after minor revision

English language and style are fine/minor spell check required.

In addition to this, the authors are kindly requested to consider citing of the following references throughout the text of their promising MS:

Nat Prod Res. 2014;28(6):372-6. doi: 10.1080/14786419.2013.869692.

Nat Prod Res. 2015;29(4):374-7. doi: 10.1080/14786419.2014.945088.

Curr Pharm Biotechnol. 2014;15(6):583-8.

Nat Prod Res. 2014;28(24):2330-3. doi: 10.1080/14786419.2014.934239.

At least in my humble opinion, this MS has a real potential to be cited quite well (in terms of hetero-citations) in the time to come.

Last but not least, very best of (research) luck ahead to all authors. 

Author Response

Reviewer 3

Well done, indeed. I really liked Your valuable manuscript (MS). My sincere congrats to all authors.

Overall Recommendation: Accept after minor revision

English language and style are fine/minor spell check required.

In addition to this, the authors are kindly requested to consider citing of the following references throughout the text of their promising MS:

- Nat Prod Res. 2014;28(6):372-6. doi: 10.1080/14786419.2013.869692.

- Nat Prod Res. 2015;29(4):374-7. doi: 10.1080/14786419.2014.945088.

- Curr Pharm Biotechnol. 2014;15(6):583-8.

- Nat Prod Res. 2014;28(24):2330-3. doi: 10.1080/14786419.2014.934239.

At least in my humble opinion, this MS has a real potential to be cited quite well (in terms of hetero-citations) in the time to come.

Last but not least, very best of (research) luck ahead to all authors.

Dear reviewer

Thank you for your very positive advice on our manuscript, we appreciate.

All changes made to the manuscript are highlighted in yellow.

Thank you also for the bibliographic references which are very interesting. In any case, the natural molecules that are described, seem promising ...

However, in our manuscript, we have voluntarily eliminated molecules that are currently undergoing clinical trials, in phase II or phase III ...

To our knowledge, we have not found evidence of toxicity tests for these molecules ... neither in animal models, nor on human cells (i.e. in vitro)... can the reviewer 3 give us more information about the progress of these molecules? in the state it seems us very delicate to introduce this type of molecules in the present manuscript.

But anyway, we tried to propose an opening towards the natural compounds at the level of the conclusion; and personally, I would like to invite the reviewer 3 or one of his colleagues to submit an article for the special issue of which I am the Guest Editor: https://www.mdpi.com/journal/molecules/special_issues/Antimicrobial_Natural_Products

Round 2

Reviewer 2 Report

Dear Authors:

While they have enriched the work and it is better presented. I believe that the conclusion must be improved and as far as possible not to place references because it seems an extension of the introduction.

Author Response

Reviewer 2

Dear Authors:

While they have enriched the work and it is better presented. I believe that the conclusion must be improved and as far as possible not to place references because it seems an extension of the introduction.

Dear reviewer

Thank you for the remarks you made, and for your time.

According to your suggestions, we have deleted the references form the conclusion, except for reference 31.

We have also modified the conclusion so that it cannot be considered as an extension of the introduction. Indeed, the FDA has just approved a brand new antibiotic, completely new. We have therefore modified the conclusion considering this new information, which is for us the proof that the search for new antibacterial molecules is indeed a hot topic.

All changes made to the manuscript are highlighted in yellow.
